# Cis-regulatory evolution spotlights species differences in the adaptive potential of gene expression plasticity

F. He[1], K. A. Steige[1], V. Kovacova[2], U. Göbel[1], M. Bouzid[1], P. D. Keightley[3], A. Beyer [1] & J. de Meaux [1]✉

Phenotypic plasticity is the variation in phenotype that a single genotype can produce in different environments and, as such, is an important component of individual fitness. However, whether the effect of new mutations, and hence evolution, depends on the direction of plasticity remains controversial. Here, we identify the cis-acting modifications that have reshaped gene expression in response to dehydration stress in three Arabidopsis species. Our study shows that the direction of effects of most cis-regulatory variants differentiating the response between *A. thaliana* and the sister species *A. lyrata* and *A. halleri* depends on the direction of pre-existing plasticity in gene expression. A comparison of the rate of cis-acting variant accumulation in each lineage indicates that the selective forces driving adaptive evolution in gene expression favors regulatory changes that magnify the stress response in *A. lyrata*. The evolutionary constraints measured on the amino-acid sequence of these genes support this interpretation. In contrast, regulatory changes that mitigate the plastic response to stress evolved more frequently in *A. halleri*. Our results demonstrate that pre-existing plasticity may be a stepping stone for adaptation, but its selective remodeling differs between lineages.

[1] CEPLAS, University of Cologne, Cologne, Germany. [2] CECAD, University of Cologne, Cologne, Germany. [3] Institute of Evolutionary Biology, University of Edinburgh, Edinburgh, UK. ✉email: jdemeaux@uni-koeln.de

Phenotypic plasticity provides most lineages with the ability to alter their development or their physiology in response to the environment[1,2]. The evolution of plastic traits has been proposed to follow one of two major models, depending on whether plasticity is adaptive or maladaptive in novel environments[3–8]. If the phenotypic state reached after activation of a plastic reaction promotes survival in a new environment, then plasticity is adaptive under these conditions. In this case, directional selection will favor mutations that move the mean phenotype towards the plastic state, via changes in the basal state or in the magnitude of the plastic response[4,9,10]. This process has been called the "Baldwin effect"[4,9,10]. This may include the selection of novel alleles that prevent the phenotype resulting from the plastic reaction from returning to its basal state. This particular "Baldwin effect" has been described as "genetic assimilation". It is predicted to help canalize the phenotype around its new optimal state, and was experimentally documented by Waddington[6–8,11]. Prominent examples have since confirmed that adaptive plasticity can provide a stepping stone for the emergence of novel genetic adaptations[12–15].

If instead the phenotype determined by a plastic reaction decreases fitness, mutations will be selected that prevent or even oppose its activation[3,16,17]. By comparing the direction of genetic change with the direction of the plastic reaction over a large number of transcripts, transcriptome studies have revealed that evolution in a novel environment often opposes the pre-existing ancestral plastic reaction[18–24]. As a result, the process of correcting maladaptive plasticity has been proposed to occur more frequently than the enhancement of plastic phenotypes via the above-mentioned "Baldwin effect"[5,18–24]. Yet, relying on expression levels alone to identify the general rules that govern how pre-existing plasticity influences the evolution of plastic reactions in a novel environment can be misleading. A single master regulator, for example, can change the expression of many genes in the same direction, even in the absence of a selective force[25]. In addition, new mutations can erode pre-existing plastic reactions, if selection is relaxed[26–33], and mutations can have functional effects whose size or direction is not uniformly distributed[31–28]. The study of allele-specific expression in F1 hybrids, instead, offers a powerful avenue to identify myriads of cis-regulatory variants that evolved independently at each transcribed locus[27,29,34–38]. With such a population of variants, it becomes possible to compare rates of evolution among lineages and distinguish the action of natural selection from the random accumulation of new regulatory mutations. The study of cis-regulatory divergence can therefore help explore the general rules governing the evolution of transcriptional plasticity.

To this aim, we investigate how cis-acting modifications have shaped the divergence in gene expression of the sister species *A. lyrata* and *A. halleri* in reaction to stress triggered by acute dehydration. The ecology of these species predicts that the reaction to dehydration stress has the most relevance in *A. lyrata*, which has the highest tolerance to dehydration[34]. In contrast, the reaction of *A. halleri* to stress is expected to be less relevant, because this species grows in highly competitive environments, and prioritizes its growth over its stress response[39]. Here, we describe how a total of 6360 and 6780 cis-acting regulatory variants identified in *A. lyrata* and *A. halleri*, respectively, contribute to reshape the transcriptome reaction to stress. Results indicate that mutations increasing expression plasticity to dehydration stress were favored in *A. lyrata*, whereas mutations decreasing stress plasticity were more frequent in *A. halleri*.

## Results and discussion

To determine how gene expression plasticity has changed between species, we monitored alterations in gene expression in *A. lyrata*, *A. halleri*, the outgroup *A. thaliana* and their F1 hybrids in response to dehydration stress. This involved sequencing the transcriptome at six time points between 0 and 24 h, replicated four times (Fig. 1A). Plants were grown under controlled conditions for four weeks and then exposed to dehydration by cutting the aerial part of the plant at the base of the root (see Supplementary Methods for details). This treatment mimicked a rupture of the water column. Our experimental design allowed us to investigate different aspects of expression changes in regards to dehydration stress. In detail, this includes the significance of gene expression variation over time (plasticity), gene expression variation across species (genetic variation) and gene expression variation as a result of time nested within species (genetic variation in plasticity). The inclusion of F1 hybrids in the experiment allowed us to simultaneously determine the contribution of cis-regulatory changes (Fig. 1B)

Compared to *A. halleri* and *A. thaliana*, we confirmed that the drought-adapted species *A. lyrata* had a higher survival rate, despite being exposed to the same level of dehydration (Supplementary Fig. 1, Supplementary Data 1, 2), and that gene expression changes overlapped with those observed in a previous study where *A. halleri* and *A. lyrata* were sampled at the onset of wilting after exposure to more progressive drought stress (Supplementary Table 1). In this species, we therefore expect that plastic reactions in gene expression will have been selected to restore homeostasis, whereas maladaptive responses that manifest as physiological and cellular stress will have been disfavored[40,41]. A principal component analysis showed that samples grouped by species and by the duration of exposure to dehydration (Supplementary Fig. 2). We detected marked differences in the stress reactions of each species, both in initial expression levels and in the slope of the response (Fig. 1 C, D), although gene expression fold changes were strongly correlated between the species pairs (Fig. 1 E–G). Among the genes whose expression was increased or decreased in response to dehydration (plastic genes), we found that, depending on the pair of species compared, between 39.5–57.9% (Supplementary Table 2) showed a significant difference in initial expression, i.e., before the onset of dehydration stress. In addition, between 22.7 and 37.4% (depending on the pair of species compared) of the plastic genes showed a significant change in plasticity itself, that is in the response slope to dehydration (false discovery rate – FDR- < 0.05, Supplementary Table 2).

**Basal expression evolution depends on the direction of plasticity.** In the vigorously growing unstressed plants sampled just before initiation of the dehydration treatment, we assume that gene expression reflects the basal expression level. The pattern of up- or down-regulation of gene expression in response to dehydration stress was strongly correlated between species pairs. After 6 h of dehydration, for example, 90% of genes that changed expression in *A. thaliana* had also changed expression in the same direction in the other two species (Fig. 1E–G). We thus assume that the direction of gene expression plasticity in *A. thaliana* in response to dehydration is a proxy for the direction of plasticity that existed in the common ancestor of the three species. If the direction in which basal expression levels evolved is independent of such ancestral plasticity, we expect that the direction of interspecific differences in expression and the direction of plasticity will be independent. To test this, we determined the genes in *A. lyrata* and *A. halleri*, whose basal expression level differed significantly from that of *A. thaliana* (there were 3262 and 4634 of these, respectively, FDR < 0.05, Supplementary Data 3). Among these, we identified those genes whose basal expression differed from *A thaliana* in the same

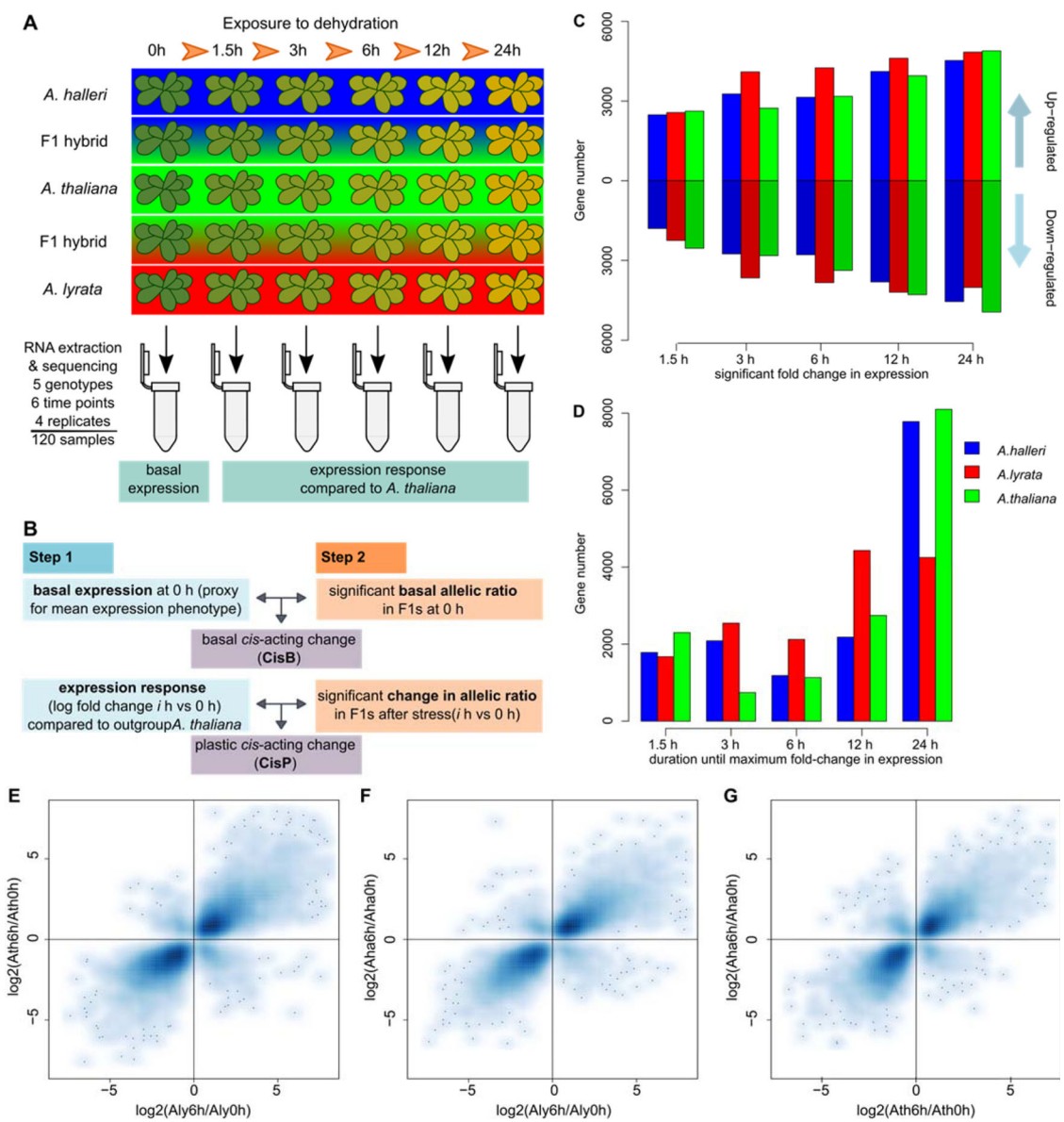

**Fig. 1 Overview of the experimental setup and number of genes responding to stress response, the time point of maximal gene expression change. A** Overview of the experimental setup for the desiccation stress for the three *Arabidopsis* species and two F1 hybrids *A. thaliana* x *A. lyrata* and *A. thaliana* x *A. halleri*. In four independent trials, aerial parts of the plants were cut and left to dry out on absorbent paper for 0, 1.5, 3, 6, 12 and 24 h under growth conditions. Samples for transcriptome sequencing were flash frozen after the treatment and used for RNA extraction. **B** Definition of basal and plastic cis-regulatory changes. If basal expression differences between the parents (step 1) can be explained by a significant basal allelic ratio (step 2) in the F1, a basal cis-acting change is contributing to the change in gene expression. If the ratio of the expression response between the parents (step 1) due to the stress can be explained by the significant change in the allelic ratio in the F1s (step 2), a plastic cis-acting change is contributing to the change in slope. Details for the analytical pipeline in Supplementary Fig. 13. **C** Number of genes showing significant up- and down-regulation at each time point after initiation of the stress in each of the three species. *A. lyrata* has a higher number of up- and down-regulated genes at intermediate time points. **D** Number of genes reaching their maximum log-fold change at each time point in each of the three species confirms the stronger response in *A. lyrata* at intermediate points. **E–G** Strong pairwise correlation in gene expression fold change between 0 and 6 h into the desiccation stress for each species pairs: **E** *A. thaliana* vs *A. lyrata* ($R = 0.84$, $p < 1e–321$), **F** *A. halleri* vs *A. lyrata* ($R = 0.85$, $p < 1e–321$) and **G** *A. halleri* vs *A. thaliana* ($R = 0.83$, $p < 1e–321$). This indicates that the overall response to the stress is similar between the species and that the direction of transcriptional plasticity predates species divergence.

direction as the plastic reaction to dehydration observed in this outgroup species and called such genetic changes in "orthoplastic" (Fig. 2A). As discussed by Crispo (2007), this term is consistent with the definition proposed by Baldwin (1896). Genetic changes in the opposite direction to the plasticity observed in *A. thaliana* we called "paraplastic" (see Fig. 2A). We note that orthoplastic basal regulatory changes are defined for a given lineage, because they are *de facto* paraplastic in the species it is compared to and *vice-versa*. This terminology thus helps describe

the evolutionary change in a given species. We observed that a majority of genes in *A. lyrata* (1975/3263, 60.5%, Fig. 2B, C) had a basal expression level that was different from *A. thaliana* in a direction orthoplastic to the direction of plasticity in *A. thaliana* (Fig. 2D, chi-squared test, $p = 2.0e–33$). When we compared basal expression levels in *A. halleri* and *A. thaliana*, we observed a similar pattern, with 64.4% (2986/4634- Fig. 2B, C, Supplementary Data 3) showing orthoplastic gene expression changes (Fig. 2D, chi-squared test, $p = 5.2e–86$). The predominantly orthoplastic

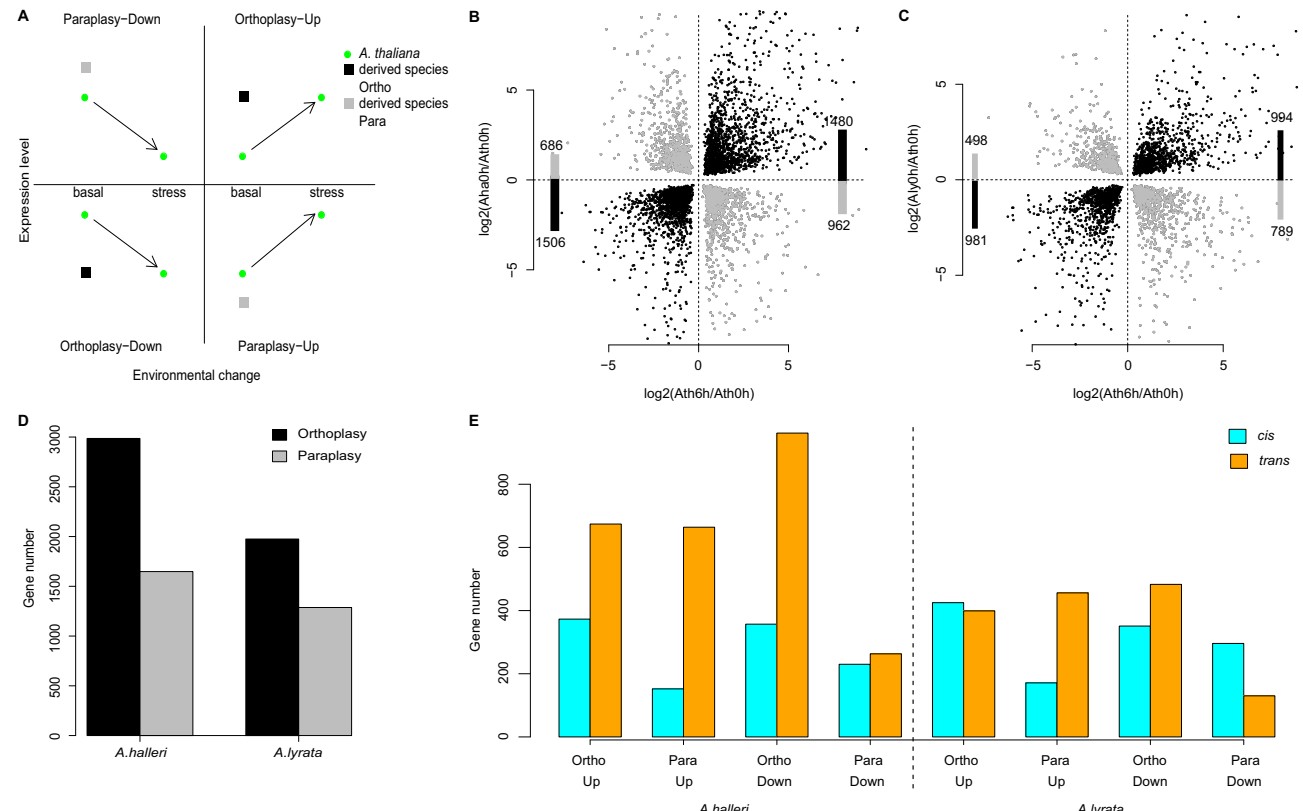

**Fig. 2 Basal expression differences between species depend on the direction of plasticity in their common outgroup species. A** Sketch illustrating what ortho- or para-plastic changes in the derived species (*A. halleri* or *A. lyrata*) correspond to for up- and down-regulated genes, using *A. thaliana* as a proxy for the plasticity and gene expression level in their common ancestor. Black dot: ortho-plastic change, gray dot: para-plastic change. Ortho- and para-plastic changes in *A. halleri* (**B**) and *A. lyrata* (**C**). *x*-axis: ratio of gene expression after 6 h in *A. thaliana*, *y*-axis: ratio of basal expression in each species, compared to expression in *A. thaliana* after 6 h. Bars represent the number in each quadrant of the plot. **D** Number of ortho- (black) and para-plastic (gray) changes in *A. halleri* and *A. thaliana*. This terminology helps describe the evolutionary change in any given species, but we note that orthoplastic basal regulatory changes in one species are de facto paraplastic in the outgroup species and vice versa. **E** The number of genes for which a basal cis-regulatory change was detected in hybrids (significant log2ratio of *A. halleri/A. thaliana* or *A. lyrata/A. thaliana* among alleles in F1 hybrid and between the corresponding parental species) or no cis-regulatory change (parental difference are not associated with a bias in allele specific expression, assumed to be controlled in *trans*).

regulatory evolution we observe when comparing *A. lyrata* or *A. halleri* to *A. thaliana* was significant at all time points (Supplementary Data 3–7), but was strongest 6 h into the stress treatment (Supplementary Figure 3A–C, chi-squared tests, minimum $p = 2.0e{-}8$).

The response in *A. thaliana* provides only a proxy for ancestral changes, and it is clear that it may differ from the response that the actual ancestor of *A. lyrata* and *A. halleri* used to deploy. Indeed, *A. thaliana* has a very different ecology and life cycle compared to the other two species and the three species diverged since more than a million years. Thus, we repeated the analysis using the reaction of *A. lyrata* or *A. halleri* for comparison with the evolution of basal gene expression in *A. halleri* and *A. lyrata*, respectively (Supplementary Fig. 4A–D). The outcome of the analysis was essentially unchanged, a result we expected because the direction of the stress response was largely conserved across species (Fig. 1E–G). We therefore conclude that, for genes responding to dehydration stress, the evolution of basal expression in the Arabidopsis genus is not independent of the direction of plasticity that likely existed before the split between *A. lyrata* and *A. halleri*.

This pattern could plausibly result from a single genetic change, if e.g. species were at different developmental stages or if they differed for only one major regulator[25]. To investigate whether this pattern resulted from more than a few trans-regulatory changes, we used F1 interspecific hybrids to determine

how many of these changes were due to cis-acting regulatory modifications. Indeed, in F1 hybrids, alleles are exposed to the same trans-regulatory environment, so that allele-specific difference in expression point to difference in cis-regulation[42]. Cis-acting variation in gene expression can be directly inferred from allele-specific expression differences observed in RNA collected from F1 hybrids between *A. thaliana* and either *A. lyrata* or *A. halleri* (Figs. 1B and 2E, Supplementary Fig. 5). The relative expression levels of the alleles within hybrids correlated positively with parental differences in expression (Supplementary Table 3, Supplementary Fig. 5). Of the 3262 genes with an altered basal expression in *A. lyrata vs. A. thaliana*, 1243 presented a significant cis-acting difference between alleles in the hybrid (FDR 0.05). A majority of these cis-acting modifications (776, 62.4%) were orthoplastic (Odds ratio 1.65, $p = 5.8E{-}10$). In *A. thaliana-A. halleri* F1 hybrids, the total number of basal cis-acting variants distinguishing *A. halleri* from *A. thaliana* was in the same order of magnitude (1112 cis-acting variants at FDR 0.05), 65.6% (730) of which contributed to orthoplastic changes in basal expression (Odds ratio 1.91, $p = 9.6E{-}14$). Therefore, cis-acting regulatory variants were predominantly orthoplastic. Because the allele-specific expression of individual transcripts reveals individual cis-acting variants, we conclude that the orthoplastic evolution of basal gene expression has a polygenic basis, both in *A. lyrata* and in *A. halleri*. The association we observed between the direction of plasticity and the evolution of basal

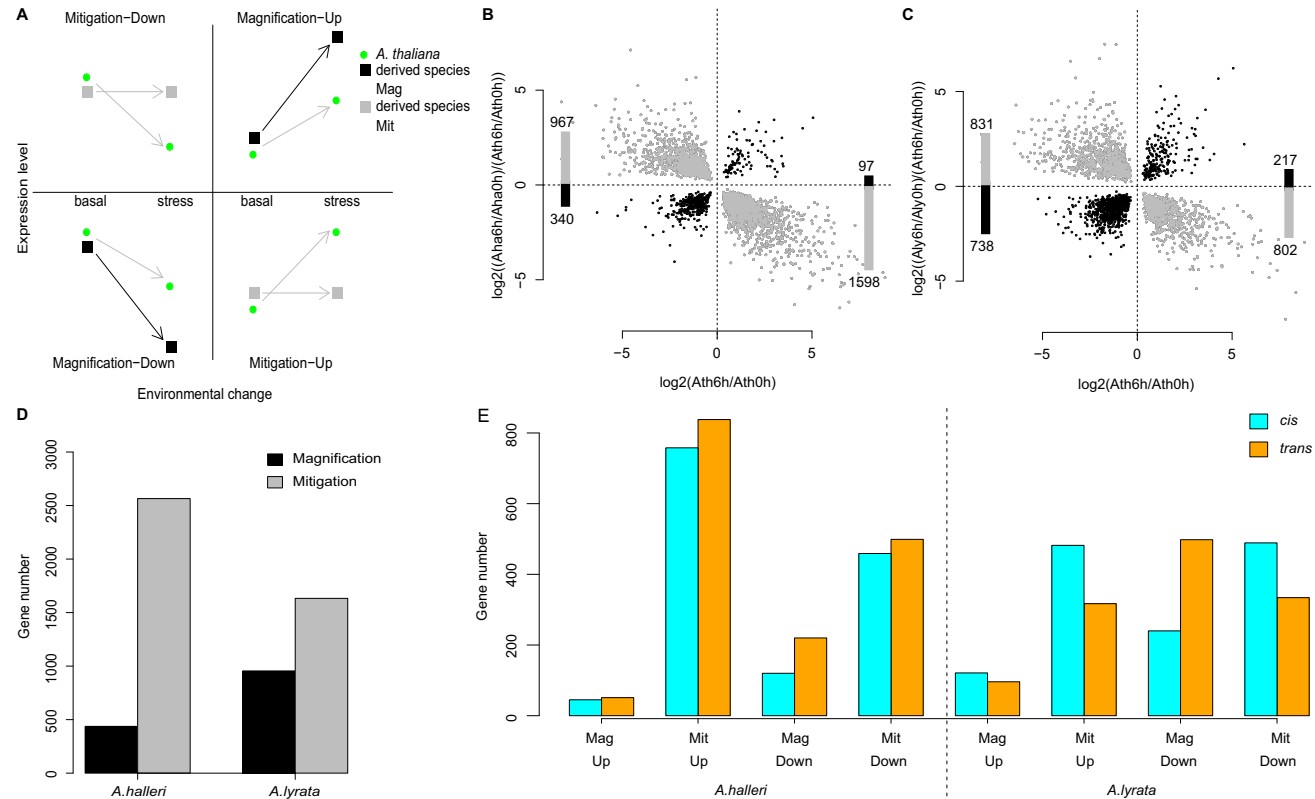

**Fig. 3 Attenuated response to the stress compared to the plasticity observed in their outgroup species. A** Sketch illustrating what magnified or mitigated response to stress in the derived species (*A. halleri* or *A.* lyrata) correspond to for up- or down-regulated genes, using *A. thaliana* as a comparison. Black arrow: magnified response, gray arrow: mitigated response. Magnified and mitigated response in *A. halleri* (**B**) and *A. lyrata* (**C**). *y*-axis: ratio of the response ratios, *x*-axis: ratio of gene expression after 6 h in *A. thaliana*. Bars represent the number in each quadrant of the plot. **D** Number of genes with a magnified (black) and mitigated (gray) response in *A. halleri* and *A. lyrata*, compared to *A. thaliana*. **E** The number of genes that have a plastic cis-regulatory change (the ratio of the parental response is explained by the log2 ratios at xh vs. 0 h for *A. halleri/A. thaliana* or *A. lyrata/A. thaliana* in the F1 hybrid) or no cis-regulatory change (the ratio of the parental response is not associated with a bias in the allele specific expression, assumed to be controlled in *trans*).

expression levels is the result of many independent regulatory mutations (Supplementary Fig. 1, Supplementary Fig. 6A).

**The response is attenuated in *A. lyrata* and *A. halleri*, compared to *A. thaliana*.** In both *A. lyrata* and *A. halleri*, we further tested whether the direction of genetic change in plasticity was independent of the direction of plasticity in the outgroup species (Fig. 3A). For this, we determined, for each time point *t* of the response, the genes whose plastic response to stress in *A. lyrata* or *A. halleri* had a slope that was significantly different from the response slope measured in *A. thaliana* (Fig. 3B–D). This revealed that for most genes, the plastic responses were attenuated in both *A. halleri* and *A. lyrata*, compared to *A. thaliana*, and, in some rare cases, even reversed (*A. lyrata*: Fig. 3B, D; and *A. lyrata*: Fig. 3C, D). The reaction to stress observed in *A. thaliana* thus tended to be predominantly mitigated in both *A. halleri* and *A. lyrata* (Supplementary Data 3). This pattern was also observed at all other time points in both species (chi-squared test, minimum *p* < 1.8E-7, Supplementary Fig. 3E–F, Supplementary Data 4–7), confirming that the effect did not result from a delayed stress reaction in one of the species[43]. Interestingly, the pattern was less pronounced in *A. lyrata* (Supplementary Fig. 3F): we observed 2.5 fold more genes in *A. lyrata* than *A. halleri* with a reaction to stress that was greater than in the outgroup *A. thaliana* after 3–6 h of stress (Fig. 3D, Supplementary Fig. 3E–H). Furthermore, for 578 of the 910 genes whose expression was down-regulated after 6 h of stress in *A. halleri*, the slope of the

stress reaction was steeper in *A. lyrata* (Supplementary Fig. 4H). The predominant evolution of a mitigated response to stress in *A. halleri* and the excess of genes showing a magnified response in *A. lyrata* were also observed if we changed the species whose reaction was used as a proxy for the magnitude of ancestral plasticity (Supplementary Fig. 4E–H).

As mentioned earlier, interspecific differences in the slope of the reaction to stress could plausibly result from differences in development or from a handful of potentially random changes in regulators of the stress response[25]. We therefore used the interspecific F1 hybrids to determine the cis-acting basis of changes in the slope of the plastic response to stress in *A. lyrata* and in *A. halleri*, compared to *A. thaliana*. If the allelic ratio measured within the hybrid were significantly altered after initiation of the stress, we concluded that a cis-acting variant had modified the response of gene expression to stress (Supplementary Fig. 5). Changes in allelic ratios within the hybrid were positively correlated with parental differences in gene expression plasticity (Supplementary Fig. 5, Supplementary Table 3). After 6 h of dehydration, a significant fold-change difference was apparent between parental species for 2588 genes, 51.4% of which were also observed in hybrids. Of these cis-acting modifications of plasticity in *A. lyrata*, a large majority (72.9%) contributed to mitigating the gene expression response after 6 h (Odds ratio 2.68, *p* < 2.2E-16). In *A. halleri*, the contribution of cis-acting modifications was also strongly skewed towards mitigation (88.1% of 1382, Fig. 3E, Odds ratio 7.37, *p* < 2.2E-16). Because the mitigated stress response observed in *A.*

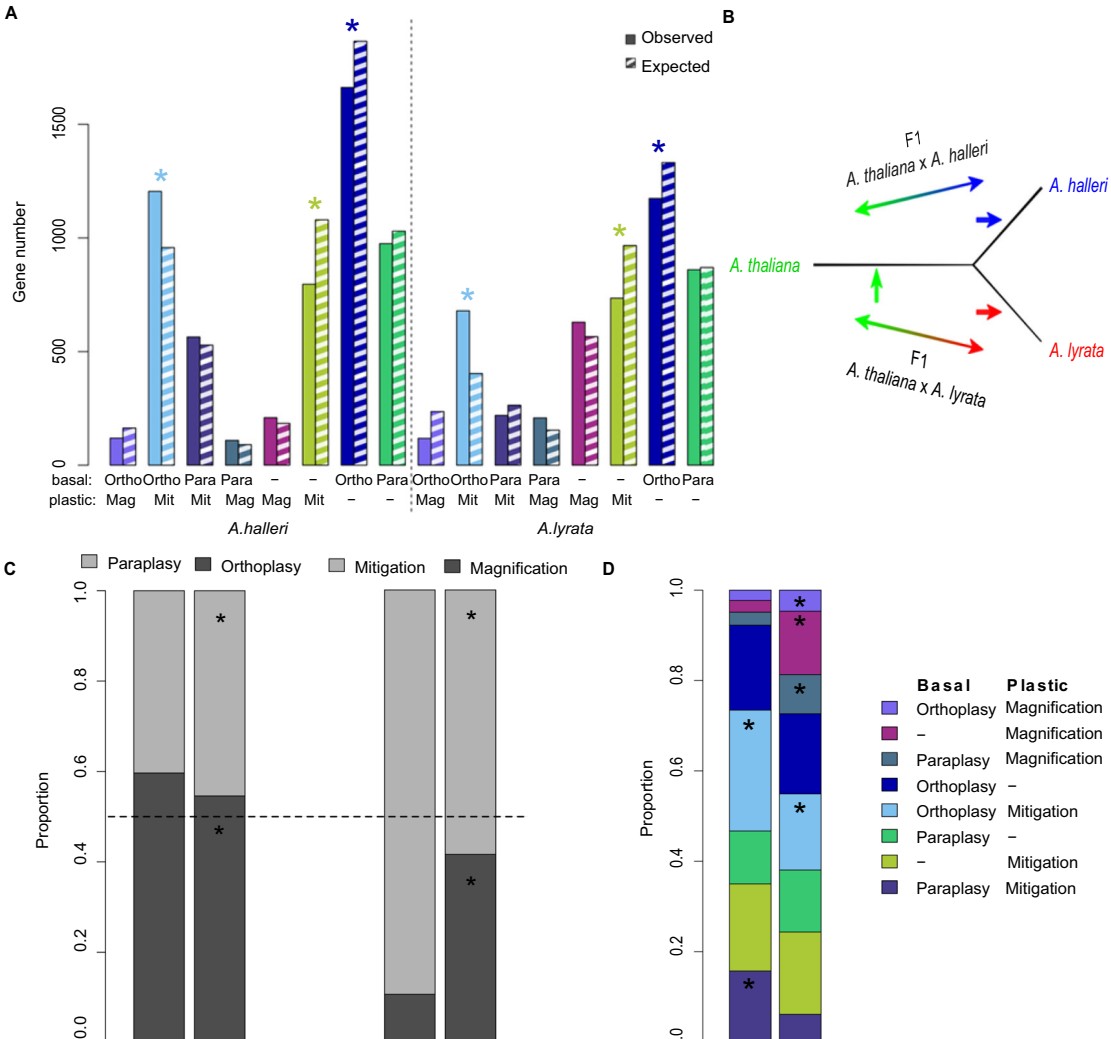

**Fig. 4 Distribution of derived cis-acting genes sets defined by their basal and/or plastic expression changes in *A.halleri* or *A. lyrata* compared to *A. thaliana*.** Ortho-Mag: orthoplasy and magnification, Ortho-Mit: orthoplasy and mitigation, Para-mit: paraplasy and mitigation, Para-Mag: paraplasy and magnification, Only-Mag: magnification only, Only-mit: Mitigation only, Only-Ortho: Orthoplasy only, Only-Para: Paraplasy only. **A** The number of observed (solid bars) and expected (hatched bars) genes in each of the 8 sets. Left: *A. halleri*; right: *A. lyrata*. "*": Significant excess or depletion, inferred by a partial Chi square test with one degree of freedom. **B** Phylogenetic relationship between the three species allowed differentiating undetermined cis-acting changes (green arrow), which are shared between the two F1 hybrids, from the derived cis-acting changes, which are specific to one hybrid (blue and red arrow). **C** Proportion of derived and cis-acting modifications within all genes with a basal change (left) or a plastic response (right). Dashed line shows 50%, i.e. the expectation in the absence of a mutational bias. The total number of genes in these groups are shown in Supplementary Fig. 6. "*": Significantly increased ot decreased number of derived changes in *A. halleri* and *A. lyrata* compared to changes predating the separation between the two species (basal: *A. halleri* vs ancestral $p = 0.09959$, *A. lyrata* vs ancestral $p = 0.0002251$; plastic: *A. halleri* vs ancestral $p = 0.4068$, *A. lyrata* vs ancestral $p < 2.2e{-}16$). **D** Proportion of derived and cis-acting modifications within each gene set. "*": Significantly increased or decreased number of derived changes in *A. halleri* compared to *A. lyrata*. The association of basal and plastic changes in gene expression is not random (Chi-sq = 47.7 df = 7 $p = 4.13e{-}08$ and Chi-sq = 302.5 df = 7 $p = 1.84e{-}61$).

*lyrata* and *A. halleri*, compared to *A. thaliana*, was clearly supported by a large number of cis-acting modifications, we conclude that it is polygenic.

**Derived cis-acting variants reveal interspecific differences in the rate of plasticity evolution.** A combination of basal expression and plasticity to stress underlies the expression profile of stress responsive genes and thus their molecular effect. Unchanged, orthoplastic or paraplastic basal expression that combine with either unchanged, magnified or mitigated plasticity define eight possible modes of plasticity evolution (Fig. 4A). A chi-squared test showed that genes were not distributed uniformly across these modes (Fig. 4A, $\chi^2 = 217.4$, df = 4, $p = 6.93e{-}46$, and $\chi^2 = 363.7$,

df = 4, $p = 1.89e{-}77$, in *A. halleri* and *A. lyrata*, respectively, Supplementary Data 3). However, to understand how the stress response evolves, we must determine how often changes in each of these modes evolve. Our experimental design allows us to compare the rate at which changes in basal expression and slope have evolved in the two sister lineages *A. lyrata* and *A. halleri*. In the absence of natural selection, we expect this rate to be similar across lineages. To distinguish derived regulatory changes in each lineage, we assumed parsimony in the evolution of cis-regulation: cis-acting variants detected in only one of the two hybrid types are more likely to have occurred after the split between *A. lyrata* and *A. halleri* (Fig. 4B, Supplementary Fig. 7). Those detected in both hybrids were called "undetermined" as it is not possible to infer

whether they predated the split of the two species or evolved in the outgroup lineage. We note that it is not possible to assign a clear direction of effect for cis-regulatory mutations of undetermined origin. The direction of effect of a derived cis-regulatory variant, instead, can be deduced by comparison with the expression level of the *A. thaliana* allele in the hybrid. We observed that in the sister species, the majority of derived cis-regulatory changes contributed to orthoplastic basal changes in expression and/or to the mitigation of the stress response observed in *A. thaliana* (Fig. 4C, Supplementary Fig. 6). Cis-regulatory variants, however, tended to remodel plasticity in different ways in *A. halleri* and *A. lyrata*. Grouping genes into the eight modes of plasticity evolution defined previously, we observed that the proportion of derived cis-acting variants in each of the two lineages was not the same across these gene groups (Fig. 4D, $\chi^2 = 189.3$, df = 7 $p = 2.02E{-}37$). The contrast was strongest after 3 h and 6 h of stress (Supplementary Fig. 3D, H, Supplementary Fig. 8). Cis-acting variants in *A. halleri* were proportionately more frequent than in *A. lyrata* among genes combining an orthoplastic basal expression change with a decrease in the slope of the response to stress (Fig. 4D, ortho-mit genes, $p < 1.42E{-}06$ after 6 h). This combination corresponds to the genetic assimilation of the plastic response observed by Waddington in his seminal experiment[4,8,11]. We also observed that *A. halleri* genes displaying a paraplastic basal expression change combined with a mitigated response had an excess of derived cis-acting variants ($p = 3.6E{-}08$). This excess indicates that regulatory modifications opposing the plastic reaction to stress were more favored in *A. halleri* than in *A. lyrata*. The over-representation, in the *A. halleri* lineage, of mutations that reduce plasticity indicates that a greater fraction of the gene expression plasticity observed in the other two species was not adaptive in this lineage.

In contrast, derived cis-acting variants were enriched in *A. lyrata* among genes displaying a magnification of the stress response, and this was true regardless of whether or not the magnification was combined with a change in basal expression (Fig. 4D, partial test $\chi^2$ tests, $p = 0.003$, $p = 1.4E{-}19$, $p = 3.9E{-}08$ for ortho-mag, mag-only and para-mag, respectively). Although a magnified response to stress was not the most frequent mode of plasticity evolution identified in this lineage, the comparatively faster rate at which cis-acting variants accumulated among these genes indicates that mutations leading to this mode of plasticity evolution are favored in the *A. lyrata* lineage compared to the *A. halleri* lineage. Interestingly, the functions enriched among them indicate that the regulation of leaf development could be one of the targets of selection for increased plasticity in *A. lyrata*. Gene Ontology categories such as GO:0010207 "Photosystem II Assembly" ($p = 1.6E{-}06$), GO:009965 "Leaf Morphogenesis" ($p = 8.4E{-}05$), GO:0010103 "Stomatal Complex Morphogenesis" ($p = 4.5E{-}06$) or GO: 0034968 "Histone Lysine Methylation" ($p = 4.5E{-}03$) were enriched (Supplementary Table 4, Supplementary Fig. 5).

**Genes with magnified plastic responses in *A. lyrata* evolve under increased evolutionary constraints**. The accumulation of derived cis-acting modifications indicates that the expression of genes displaying magnified plasticity after 6 h of stress evolves under positive selection in *A. lyrata*. If such genes with magnified plasticity are subject to stronger selection overall in this lineage, we also might expect that the genes will be exposed to stronger evolutionary constraints than other plastic genes. We grouped *A. lyrata* genes according to their mode of plasticity and assessed the ratio of the divergence at non-synonymous and synonymous sites ($Ka/Ks$) between *A. lyrata* and *A. thaliana*. We additionally included genes that did not change their expression as a control group and used bootstrapping to determine confidence intervals and identify

significant differences between groups. For three of the eight modes of regulatory evolution, the number of genes was small, resulting in high bootstrap variance (Fig. 4A, Supplementary Fig. 9); these modes were therefore excluded from the analysis. We observed that genes of all remaining modes of plasticity evolution showed a significantly lower $Ka/Ks$ than the control group (Fig. 5A). Genes with modified plasticity therefore all experienced stronger evolutionary constraints. Interestingly, however, we observed that the $Ks$ of control genes was significantly lower than the $Ks$ of all gene groups, with the exception of the genes that evolved a magnified response to stress (Fig. 5B, C, Supplementary Fig. 9H–J). To gain further insight into the evolutionary forces operating on these genes in *A. lyrata*, we examined nucleotide polymorphism within a local *A. lyrata* population[44–46]. The average number of pairwise differences did not differ significantly across groups of genes, and was similar to that of the control genes (Fig. 5D). We also used the Tajima's $D$ statistic to quantify the contribution of low-frequency variants to nucleotide diversity within each gene group. Mean $D$ for non-synonymous sites differed significantly among modes of plasticity evolution (Fig. 5E), but mean $D$ did not differ significantly for synonymous sites (Fig. 5F). *A. lyrata* genes that evolved a magnified plastic reaction, but no basal change in expression, displayed the most negative Tajima's $D$ of all modes of plasticity evolution at non-synonymous sites (Figure 5E, F). We performed the same analysis on the polymorphism for orthologous gene sets within *A. thaliana*. Compared to *A. lyrata*, Tajima's $D$ in *A. thaliana* was generally lower, presumably due to its history of population expansion[45]. Yet, the distinctive Tajima's $D$ of *A. lyrata* genes with a magnified plasticity was specific to *A. lyrata*, because the Tajima's $D$ of its orthologous *A. thaliana* genes was not lower than that of orthologous genes in the other plasticity groups (Fig. 5A, G). This finding indicates that amino-acid variants in *A. lyrata* tend to segregate at lower frequencies, if they affect genes with a magnified plasticity (Fig. 5G). To confirm that the amino-acid sequence of genes with a magnified plasticity are subject to stronger evolutionary constraints, we estimated the distribution of fitness effects (DFE) of new non synonymous mutations for each gene group, using the software fitδaδi[47,48]. Parameters of a demographic model for the *A. lyrata* population have been previously determined[46]. Population data, simulated under this model by fitδaδi, fitted well with the site frequency spectrum of synonymous variation observed in the population (Supplementary Fig. 10). The DFE was then subsequently estimated by fitting a gamma distribution of fitness effects to the frequency spectrum of non-synonymous variants[47] (Fig. 5H). This analysis showed that, of all modes of plasticity evolution, the genes with a magnified plastic reaction to stress showed the lowest proportion of variants with weak fitness effects. Bootstrapping confirmed that the predicted proportion of nearly neutral deleterious variants was significantly lower among these genes (Supplementary Fig. 11). This pattern is in agreement with the higher proportion of low-frequency variants indicated by the lower Tajima's $D$ statistic of this group of genes, compared to other plastic genes (Fig. 5H, Supplementary Fig. 11). In conclusion, *A. lyrata* genes that evolved a magnified response after 6 h of stress not only accumulated an excess of derived cis-acting variants, they also appear to be subject to increased selective constraints at the amino-acid level.

Waddington proposed that stabilizing selection around the new optimum (canalization) would favor the genetic assimilation of plastic responses[11]. In *A. lyrata*, the constraints detectable on genes with an assimilated stress response that combine the two predominant types of basal and plastic regulatory changes have neither increased nor decreased (Fig. 5A–D). The fitness effects of new mutations in these genes do not differ significantly from those in genes whose regulation remained unchanged (Fig. 5A–D, Supplementary Fig. 9). This result indicates that the genetic assimilation of the ancestral plastic response is unlikely to

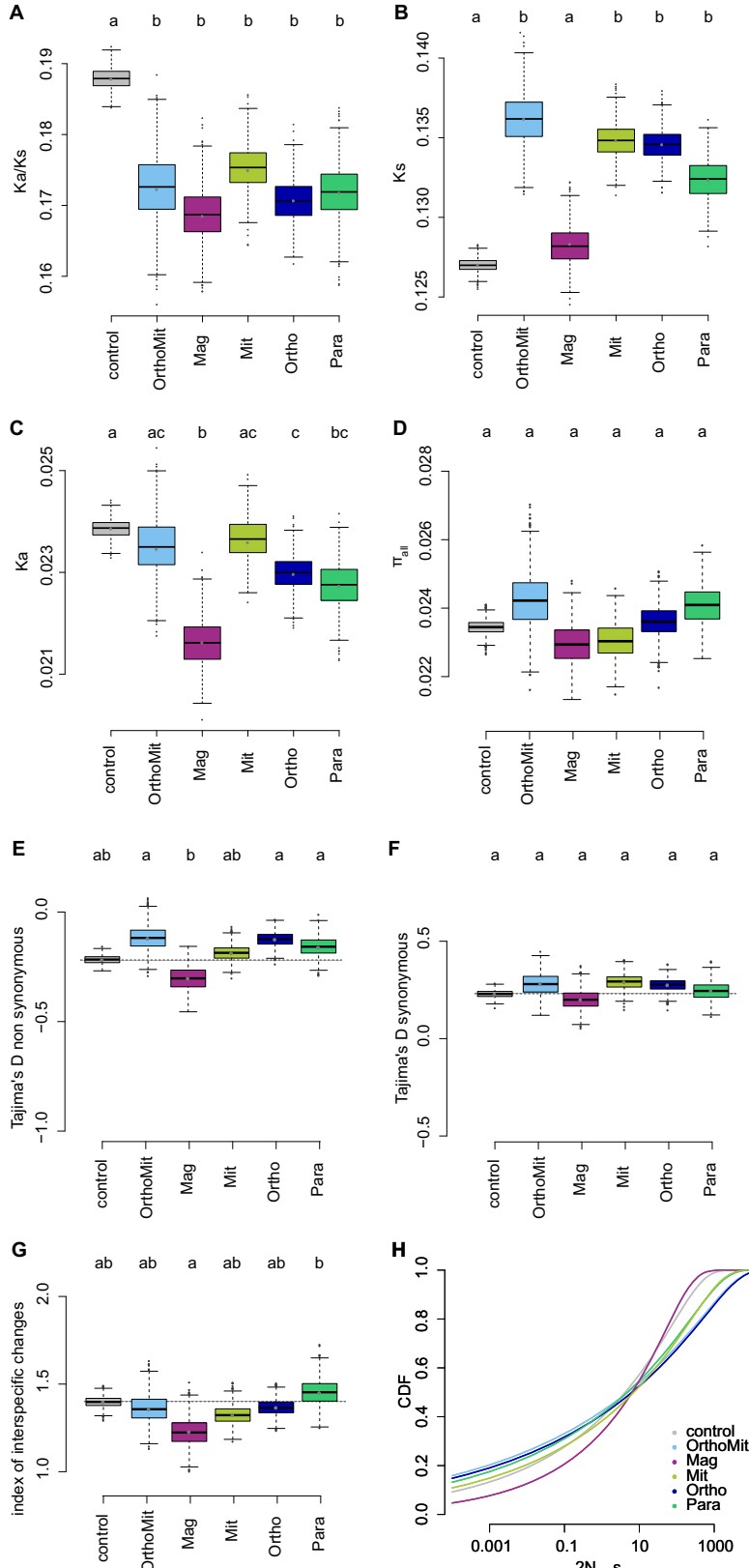

associate with increased canalization at the amino-acid level in *A. lyrata*. Future studies will have to examine whether the excess of cis-acting changes leading to a decrease in plasticity in *A. halleri* reflects selection to lower the energetic cost associated with

plasticity or whether it is a result of relaxed constraints on the regulation of the stress response[49].

The rapid and complex plastic responses plants have evolved allow them to tune reactions to ever-changing environmental

**Fig. 5 Tajima's D at synonymous and non-synonymous sites differ between 5 modes of plasticity evolution.** Ortho-Mit: orthoplasy and mitigation, Only-Mag: magnification only, Only-mit: Mitigation only, Only-Ortho: Orthoplasy only, Only-Para: Paraplasy only. The plasticity categories Ortho-Mag, Para-Mit and Para-Mag were excluded due to having too few genes (see Fig. 4A) and therefore too high variance. **A** Ka/Ks ratio, **B** Ks between *A. lyrata* and *A. thaliana*, **C** Ka between *A. lyrata* and *A. thaliana*, **D** average π per gene, **E** Tajima's D for synonymous sites and **F** nonsynonymous sites in *A. lyrata*. Significance between the groups was estimated based on pairwise comparison of 1000 bootstrap replicates, by using the union of the bootstrap values to calculate what proportion of the differences between the groups significantly differed from 0, multiplied by 2, to account for the fact that the test is one-sided. Box and whiskers depict the 75th and 95th interquantile ranges, respectively, dots show outliers. No shared letters mean that the groups are different with a significance cut-off at $p < 0.05$. P-values of all pairwise comparisons can be found in Supplementary Data 8. **G** Relative change in mean Tajima's D for non-synonymous sites between *A. lyrata* and *A. thaliana*. (($A. lyrata + 2$) / ($A. thaliana + 2$)) and **H** Cumulative distribution function of the gamma distribution for the control and the 5 classes of plasticity. OrthoMag, ParaMit and ParaMag were removed due to the low number of genes in these plasticity classes.

stressors, with ecological priorities for survival, growth and reproduction[50]. By documenting the cis-regulatory divergence between *A. halleri* or *A. lyrata* and their common outgroup *A. thaliana*, we show that the divergence of plastic stress responses in the *Arabidopsis* genus has a broad polygenic basis. By comparing the direction of cis-acting changes with the direction of the plastic reaction, we demonstrate the profound impact of plasticity on the accumulation of regulatory mutations in each lineage, lending the strongest support to date for the idea that plasticity can be stepping stone for the evolution of transcription[7]. Previous studies concluded that most transcript changes in reaction to stress were maladaptive because they were predominantly lost during adaptation[18,20–22]. Here, we challenge these approaches by using two sources of information to infer the adaptive relevance of regulatory changes in the plastic response to stress. First, by comparing allele-specific expression across the two interspecific hybrids, we estimated lineage-specific rates of accumulation of derived cis-acting variants, an approach that has proven successful in revealing the polygenic response to selection on complex traits[29,48]. Second, we used the site frequency spectrum of synonymous and non-synonymous mutations in *A. lyrata* to test whether the fitness effects of new mutations in plastic genes depends on the mode of plasticity evolution[29]. In *A. lyrata*, the lineage that best survived the severe dehydration imposed in this experiment (Supplementary Fig. 1) and displays the strongest drought tolerance[39,51], both approaches point to the sustained action of polygenic selection for an increase in expression plasticity of many genes, rather than its loss. In *A. halleri*, instead, we did observe an excess of cis-acting variants leading to a decreased transcriptional plasticity to stress. We conclude that plasticity can potentiate the adaptation of gene expression, but its selective remodeling depends on the ecological context of each lineage.

## Methods

**Plant materials and stress treatments**. Seeds of *A. thaliana* accession Col-0 and *Arabidopsis lyrata* ssp. *lyrata* genotype MN47 were obtained from the Arabidopsis Biological Resource Center (ABRC, USA). Seeds of *A. halleri* h2-2 (Gorno, Italy) were obtained from Pierre Saumitou-Laprade (University of Lille, France). Based on previous work, these genotypes are representative of the difference in drought tolerance between the three species[39]. F1 crosses were generated by pollinating emasculated *A. thaliana* flowers with pollen of *A. lyrata* (AthxAly) or *A. halleri* (AthxAhal), as described in de Meaux et al.[52]. Crosses with *A. thaliana* as the pollen parent were unsuccessful and thus no reciprocal F1s were included.

Col-0, MN47, AthxAly and AthxAha F1 hybrids were germinated and grown on germination medium containing Murashige and Skoog salts, 1% sucrose, and 0.8% agar. The plants were stratified for 3 days at 4 °C, and then transferred to soil. Plants were grown for 4 weeks. *A. halleri* plants were multiplied by clonal amplification and grown for three weeks. Plants were all randomly placed and grown in a chamber at 20 °C under 14 h light/16 °C 10 h dark under dim light (100 mmol sec±1 m ± 2). Dehydration treatment was applied in four independent trials performed at one-week intervals and followed a published protocol[53]. Germination or cuttings were staggered over four weeks to ensure that plant size at harvest was comparable across the four trials. For the dehydration treatment, the aerial part of the plants was cut at the base of the roots to mimic a rupture of the water column, as occurs when the plant wilts, and deposited on absorbent paper for 0, 1.5, 3, 6, 12

and 24 h in the growth chamber under the same growth conditions. Plants were weighed before and after the stress prior to flash freezing for RNA extraction to verify that the rate of water loss was similar for all three species (Supplementary Fig. 1). The ecological relevance of this treatment was confirmed in a second experiment by determining the different survival rates of the plants in response to the stress (Supplementary Fig. 1). *A. lyrata* was the only species to survive 24 h of acute dehydration, a result in line with its known ability to withstand several days of wilting[39]. Additionally, there was a significant overlap of stress-responsive genes in this experiment to the stress-responsive genes in Bouzid et al[39]. (hypergeometric test, Supplementary Table 1), where the authors performed a more progressive dry-down experiment conducted in potted plants.

**RNA isolation, preparation of cDNA libraries and sequencing**. The whole aerial part of parental or hybrid individuals was flash frozen in Eppendorf tubes using liquid nitrogen and homogenized to fine powder in a Homogenizer of Precellys Evolution (Bertin Technologies). Total RNA was extracted in 1 ml of Invitrogen TRIzol Plus RNA Purification System (Thermo Fisher Scientific) and decontaminated with the DNA-free kit (Thermo Fisher Scientific). RNA quality and quantity were examined with the Bioanalyzer 2100 (Agilent) and Qubit® 2.0 Fluorometer (Thermo Fisher Scientific). Two microgram of Total RNA was used for library preparation. Library preparation followed the TruSeq® Illumina RNA Sample Preparation v2 Guide. Sequencing was performed on Illumina HiSeq2000 following the manufacturer's protocols, and paired-end 100 bp long reads were obtained. 15.6-21.5 million paired reads for each sample of the parents and 33.4-41.1 million reads for each sample of the hybrids were produced.

**Read processing**. After FastQC quality check (www.bioinformatics.babraham.ac.uk/projects/fastqc/), the FastX-toolkit was used for sequence trimming and filtering. Low-quality nucleotides were removed from the 3′ ends of the sequences (fastq_quality_trimmer -Q 33 -t20 -l 50). Sequences were reverse-complemented by fastx_reverse_complement, and 5′ ends were cut following the parameter described above, before being reverse-complemented back to the original direction of reads. Reads with less than 90% bases above quality threshold (fastq_quality_filter -Q 33 -q 20 -p 90), and paired-end reads with a single valid end were removed by a custom R script. Trimmed and filtered reads were saved for further analysis.

For *A. halleri*, we generated a pseudo-genome using the *A. lyrata* genome and an assembly of the *A. halleri* transcriptome. For this, the *A. halleri* transcriptome was first assembled as described below using the bulk of 24 *A. halleri* transcriptomes yielding a total of 456 million paired-end reads generated in this study. After verifying assembly quality (91% transcripts are single-copy orthologues), we used the assembly to generate a pseudo-reference genome for *A. halleri*.

The *A. halleri* pseudo-reference genome was generated by aligning the *A. halleri* transcriptome to the *A. lyrata* genome with BLAT[54], calling the SNP variants after the alignment and substituting them on the *A. lyrata* genome with a custom script. The proportion of mapped reads for samples of *A. halleri* and its hybrid using *A. halleri* pseudo genome was similar to the proportion of mapped reads for samples of *A. lyrata* and its hybrid obtained using *A. lyrata* genome as a reference (see Supplementary Table 6, Supplementary Fig. 12). When *A. lyrata* is used as a reference for *A. halleri* transcriptome, ~70% of the total reads map to the reference. However, 90% of the *A. halleri* reads can be uniquely mapped on the *A. halleri* pseudogenome, a percentage of mapped reads comparable to the percentage obtained for *A. lyrata* reads mapped to the *A. lyrata* reference (~90%). For the hybrids, using a hybrid pseudo-reference genome based on the two parental species increased read mapping from ~80 to ~90% for both the *A. thaliana x A. halleri* and *A. thaliana x A. lyrata* transcriptome, decreased mapping bias and improved the quantification of allele-specific expression (Supplementary Fig. 12C and below).

**A. halleri transcriptome assembly**. An assembly of the transcriptome of *A. halleri* was generated using a total of 456 million paired-end reads generated from all A. halleri transcriptome samples during the desiccation time-course. For this, transcriptome reads were first concatenated into two groups (*_left.fq > all_left_reads.

fq; *_right.fq > all_right_reads.fq). These reads were then mapped to the rDNA/ TEs/REs plant databases SILVA (https://www.arb-silva.de) and MIPS PlantsDB[55] using STAR version 2.5.0c[56]. All reads mapping to rRNA and TEs were removed using SAMtools version 1.8 (samtools fastq -f 4, Li 2011 Bioinformatics) and read errors were corrected using CORAL version 1.4.1[57]. We further removed low quality reads and unpaired reads using TRIMMOMATIC version 0.36[58]. For the remaining reads, we normalized read counts using the trinity-in-silico-normalize function and then preformed a de novo assembly using TRINITY version 2.2.0[59]. The assembly was then filtered based on IsoPct and length (the longest isoform with the highest IsoPct) using RSEM version 1.3.0[60] and duplicated contigs were removed using CAP3[61]. Using BUSCO version 1.2[62], we confirmed the high quality of the assembly we obtained (91% single-copy).

**Parental and hybrid gene expression**. The complete analytical pipeline is summarized in Fig. 1A, B and Supplementary Fig. 13. For *A. thaliana*, *A. lyrata* and *A. halleri* samples, all the reads were mapped to their own genomes using STAR with default parameters[56]. Read count numbers were calculated by HTSeq-count[63] and differential gene expression was analyzed using DESeq2[64] with a nested model (Expression ~ Species/time). Read counts were computed as number of fragments per kilobase per million reads (FPKM). Variance in standardized read counts did not increase with the duration of stress, thus indicating that the reaction to acute stress remained tightly controlled throughout the stress treatment (Supplementary Fig. 11). A Spearman correlation coefficient was computed for each gene, based on gene rank for read count, to quantify the among sample correlation in expression level reported in Fig. 1. The significance of gene expression variation over time (plasticity), gene expression variation across species (genetic variation) and gene expression variation as a result of time nested within species (genetic variation in plasticity) was tested with a generalized linear model implemented in DESeq with read count as dependent variable and time nested within species as independent variable (note that DESeq normalizes read counts across samples). Gene expression differences at 0 h (e.g., just before initiation of the stress) were assumed to be constitutive and reflect basal expression level (Supplementary Data 3–7). Gene expression plasticity was quantified for each time point $t$ of the time series as the slope of the response to stress after $t$ hours. For this we computed the ratio of gene expression at time point $t$ divided by gene expression at time point $t_0 = 0$ h. The significance of the plastic response was determined with the CONTRAST feature of the RESULTS function of DESeq2[64]. Differences in plasticity between species were further computed as the log2ratio of gene expression plasticity measured in each species, that is the ratio of allele count after $t$ h over allele count at $t_0 = 0$ h in the first species divided by the ratio of allele count after $t$ h over allele count at $t_0 = 0$ h in the second species (e.g., log2 $[(Aly_{t=6h}/ Aly_{t=oh})/ (Ath_{t=6h}/ Ath_{t=oh})]$ - Supplementary Data 3–7). A negative value of this ratio indicates either a weaker response to the stress or a reversal in the direction of the plastic change, where for example a gene would be up-regulated in one species and down-regulated in the other species. The latter case, however, was rare. More than 90% of the genes that responded to stress in *A. thaliana*, responded in the same direction in the other two species (Fig. 1E, G). In *A. lyrata* and *A. halleri*, genes with species-specific differences in plasticity overlapped significantly with plasticity differences reported after exposing plants to progressive soil dehydration[39].

To define the genetic basis of gene expression change, we determined allele-specific expression levels within the F1 hybrids obtained from crossing each of the two sister species *A. lyrata* and *A. halleri* with their outgroup relative *A. thaliana* as first described in de Meaux et al.[52] (Fig. 1A, B, Supplementary Fig. 13). Briefly, allelic read counts were determined by mapping the F1 hybrid transcriptome to a hybrid genome concatenating the *A. thaliana* Col-0 reference genome (TAIR10, www.arabidopsis.org) with either the *A. lyrata* MN47 reference genome (Araly1[65]) or with the *A. halleri* pseudogenome. The trimmed and filtered reads were mapped to the hybrid genome using STAR version 2.5.0c[55] with the built-in Bowtie2 mapping program[66]. Uniquely and high-quality mapping reads were selected by "samtools view -q 10" (-b: output is bam files, -q 10, mapping quality of phred score that means 90% possibility of the mapping is correct). We focused our analysis on orthologs present in each of the three species, *A. thaliana*, A. lyrata and *A. halleri*. The Arsly1 genome annotation detects 17846 orthologous genes in *A. thaliana* and *A.lyrata*. The sequences of these orthologs were blasted against the *A. halleri* draft genome sequences of Aha1.1 (Phytozome 10.1, Arabidopsis halleri v1.1). Criteria for orthologous genes was set as hit E-value < 1 e -20) and pairs of genes with reciprocal best BLAST hits were defined to be orthologs. A total of 528 genes had no detectable orthologs in the *A. halleri* draft genome, resulting in a set of 17 318 orthologous genes in the three species.

We added several steps to prevent distortion due to mapping bias. First, unmapped *A. halleri* reads were cut in 30 bp-long read fragments and re-mapped to the reference of *A. lyrata*. Second, we used a set of SNPs fixed in the two sister lineages to re-assign reads that had maps reads on the wrong parental genome, because the bowtie2 algorithm sometimes maps reads on the wrong parental allele. For this, the orthologous cDNA sequences of *A. thaliana* and *A. lyrata* were aligned using MAFFT[67] and 1.34 million SNPs were located. A collection of *A. halleri* derived SNPs was provided by P. Novikova (MPI-PZ, Cologne). This collection was used to select the 1.02 M SNPs that were fixed in the common ancestor of *A. lyrata* and *A. halleri*. This set of SNPs was used to re-assign the parental origin of each read, based on Samtools SNP callings. We also observed that regions close to intron and/

or highly divergent segments, often failed to map properly, causing a mapping bias. We therefore quantified relative allelic abundance focusing on SNPs more than 50 bp away from introns, excluding regions with more than 10 SNPs in 200 bp (greater than the mapping parameter of mismatch number 5 in 100 bp) to minimize mapping bias caused by highly divergent (and/or misannotated) gene regions. Loci with a mean final read count less than 10 were discarded from the analysis to exclude low-expressed genes. Finally, we excluded any SNP position for which the total coverage in the DNA samples was less than 5 reads. After all these filters, 252454 SNPs were kept here (14.6 SNPs per gene). Final parental read counts at each SNP included both mapped reads and remapped 30 bp fragments, and the median allele ratio SNP ratio was computed for each gene. Finally, each parental read count was divided by a size factor (Total read count for each ortholog/ median total read counts for all orthologs[68]). We used previously published DNAseq data for the hybrids[29] to confirm that these filters allowed an effective control of mapping biases.

We then identified genes showing significant allele-specific expression (ASE) with a General Linearized Model (GLM) including a quasi-binomial distribution of error. Under this model, the null hypothesis $H_0$ assumes no difference between RNA and DNA samples (Allele ratio of $\mu_{RNA} = \mu_{DNA}$), the alternative hypothesis $H_1$ assumes ASE at the gene considered (ASE, Allele ratio of $\mu_{RNA} \neq \mu_{DNA}$). In other words, the null hypothesis $H_0$ assumes no differences in the allele ratio between RNA and DNA reads, whereas the alternative $H_1$ assumes ASE. The GLM model used the read counts of each of the two alleles as dependent variable, and tested for the effect of the 7 sample types (DNA, RNA 0 h, RNA 1.5 h, RNA 3 h, RNA 6 h, RNA 12 h and RNA 24 h) nested within each hybrid type. Loci with a mean final read count < 10 were discarded from the analysis to exclude low-expressed genes. Significant ASE genes were defined by contrasting treatment specific allelic ratios to DNA allelic ratios. P-values were adjusted for controlling the false discovery rate (FDR correction[69]). An FDR threshold of 5% was fixed to call significant ASE genes.

**Inferring cis-acting contribution to parental differences**. We analyzed separately constitutive expression difference (i.e., allele or parental expression changes measured at time point 0, before initiation of the stress), and differences in expression plasticity, to account for interspecific differences in the timing and magnitude of gene activation or repression (Figs. 1B, 2A and 3A, Supplementary Fig. 13). Low expression genes in either parent or hybrid (mean count <10) were removed from the analysis.

Comparing parental and allele-specific expression differences, we defined three categories of gene expression changes, akin to the categories defined by Wittkopp et al.[42]. We first defined basal expression differences and their genetic architecture (Supplementary Fig. 13, Supplementary Fig. 5). If parental genotypes differed significantly in expression (FDR < 0.05) while no significant difference in allele-specific expression (ASE) was detected in their hybrid (FDR > 0.05), the genetic basis of the expression change was defined as trans-only change. If significant differences were detected both between parental genotypes (FDR > 0.05) and between alleles in the hybrid (FDR > 0.05), a cis-acting variant was inferred to contribute to the expression change. We did not consider genes where ASE was detected in the absence of a significant parental difference (Supplementary Fig. 5, purple). Indeed, since alleles are expressed in the same environment in F1s, we have greater power to detect allele specific differences than differences between parents. We also did not consider the (rare) genes where the preferred expressed alleles in the hybrid did not belong to the parent expressing the gene at the highest level (Supplementary Fig. 5).

We then determined genes with a plastic cis-acting variant as the genes for which a change in allele-specific expression contributed to change in the plastic response after initiation of the stress (Supplementary Fig. 5, Supplementary Fig. 13, Supplementary Table 3). For the analysis of variation in plasticity of gene expression during stress, we used the ratio of plastic expression after 1.5–24 h of stress over gene expression at 0 h. Significant difference was determined in a single GLM model with quasi-binomial distribution nesting the duration of stress within hybrid type or parental difference. For the hybrids, we used the ratio of the ASE ratio at 1.5–24 h of stress vs the ratio of ASE at time point 0 h (tp 0, FDR < 0.1). Genetic variants of expression plasticity in response to stress were called significant at FDR < 0.1 (Supplementary Fig. 5 – xh vs 0 h).

**Testing for the impact of plasticity on the direction of gene expression evolution**. *A. thaliana* genes with a significantly higher or lower expression after 1.5–24 h (FDR < 0.05) were defined as plastic and partitioned into up-regulated or down-regulated genes. To test for the impact of the direction of plasticity on the direction of evolutionary change in basal expression level, we used the plastic response of *A. thaliana* as a proxy for the ancestral direction of plastic stress response, since this species forms an outgroup to the sister species *A. halleri* and *A. lyrata*. We then determined whether changes in basal expression in each species were orthoplastic (i.e., in the direction of supposed ancestral plasticity) or paraplastic (i.e., in the direction opposite to supposed ancestral plasticity).

For each gene, we plotted the ratio of expression level of *A. lyrata* or *A. halleri* over the fold change in expression observed in *A. thaliana* (e.g., interspecific evolutionary change in expression significant at FDR 0.05) as a function of the plastic response in *A. thaliana* (significant stress response at FDR = 0.05) (Fig. 2A).

We then counted the number of genes whose interspecific difference in basal expression (at 0 h, before initiation of the stress) was in the direction of the stress response in *A. thaliana* (orthoplastic) or opposite (paraplastic). We subsequently used a chi-squared test with 1 degree of freedom to test whether gene counts were randomly distributed in the 4 quadrants of the plots, thereby estimating the proportion of genes evolving by orthoplasy, given their probability to be up- or down-regulated by stress (Fig. 2A–C). We subsequently used a chi-squared test with 1 degree of freedom to test whether the proportion of genes with a detectable cis-acting variant were randomly distributed among these genes. Finally, we compared the distribution in each quadrant in *A. lyrata* with the distribution expected based on the distribution observed in *A. halleri*, and used a chi-squared test with 4 degrees of freedom to determine significant differences in the contribution of cis- acting variants to the modification of basal expression levels.

A similar approach was used to test for the impact of the direction of plasticity on the direction of evolutionary change in plasticity, which is, as already described above, the ratio of allele count after $t$ h over allele count at $t_0 = 0$ h in the first species divided by the ratio of allele count after $t$ h over allele count at $t_0 = 0$ h in the second species. In other term, the change in plasticity was computed as log $[(Aly_{t=6h}/ Aly_{t=oh}) / (Ath_{t=6h}/ Ath_{t=oh})]$. A positive value indicates an increase in the magnitude of the plastic response (magnification) and a negative value indicates a decrease in magnitude and/or a reversion of the plastic reaction (mitigation). Note, however, that for 90% of the genes, the direction of gene expression change was conserved between species. Significant plasticity changes were taken at FDR 0.1 (Fig. 2A–C). In this way, we computed the proportion of genes evolving by a magnified vs. mitigated plastic response as well as the contribution of cis-acting variants to these evolutionary outcomes (Supplementary Fig. 3G, H, Supplementary Fig. 5).

**Comparative analysis of the distribution of fitness effects in *A. lyrata* and *A. thaliana*.** For analyzing the strength of selection, whole genome data for 16 *A. lyrata* individuals collected in Plech (Germany) were taken from Takou et al.[46], and 40 *A. thaliana* individuals from Spain were taken from 1001 genomes[70]. Individuals were randomly down-sampled to have the same number of observed alleles for all sites. Vcf files were filtered to remove Indels, sites with more than two alleles, sites with coverage <10, genotype quality <20 or quality <30 as well as positions with more than 80% of the individuals missing data, as described in Takou et al.[46]. We focused on orthologous genes in this dataset that had polymorphism information in both species. Synonymous and nonsynonymous positions were retained and folded site frequency spectra were generated for non-overlapping groups of genes defined for each mode of plasticity evolution using R (Supplementary Fig. 10). As an additional control, we included a group of plastic genes, whose plastic reaction did not change between *A. lyrata* and *A. thaliana*. Additionally, summary statistics ($\pi_n/\pi_s$, Tajima's D, $\pi$) were calculated using custom R scripts for each of the categories. Ka and Ks were calculated using an alignment of *A. lyrata* and *A. thaliana* as described in He et al.[29], with the "kaks" function of R package seqinr. For each summary statistic 1000 bootstrap replicates were generated by resampling the genes with replacement, and recalculating the statistics over all genes. An index of interspecific changes was calculated by the ratio between the species $((D_{Aly} + 2)/ (D_{Ath} + 2))$ for the bootstrap replicates. Differences between plasticity groups were assessed by calculating if the difference between the groups significantly differed from 0 for each bootstrap by using the union of the bootstrap values to calculate what proportion of the differences between the groups significantly differed from 0, multiplied by 2, to account for the fact that the test is one-sided. Note that for three mode of plasticity evolution (paraplasy-mitigation, orthoplasy-magnification, paraplasy-magnification), the number of genes was small and bootstraps revealed high variance. Those modes of plasticity evolution were therefore not further considered in the analysis.

The program fitδaδi[47] was used to estimate the distribution of fitness effects. This extension to the δaδi program[71], which infers demographic history and infers selection based on genomic data, allows us to specify the demographic model when inferring selection. In short, we are using the simplified demographic model for the *A. lyrata* Plech population based on the demographic model in Takou et al.[46]. The synonymous population scaled mutation rate was estimated using the δaδi function Inference.optimal_sfs_scaling and multiplied by 2.76 to get the nonsynonymous mutation rate. The nonsynonymous SFS was then used to estimate the selection parameters which are given by the shape and scale parameter of the gamma distribution. 200 bootstrap replicates of the resampled SFS were used to get 95% confidence intervals for the gamma distribution parameters.

**Inferring the phylogenetic origin of cis-acting variants and their lineage-specific rate of accumulation.** To infer the phylogenetic origin of cis-acting modifications, we defined *cis* acting changes as ancestral if they were shared between the two types of F1 hybrids, which shared *A. thaliana* as an outgroup. If the cis-acting variant was specific to one of the hybrids, this change was considered to be derived in the corresponding parent, i.e., *A. lyrata*- or *A. halleri*-derived. Genes that displayed a change in plasticity contributed by at least one cis-acting modification (e.g., basal cis-change, plastic cis-change or both) were considered as independent variants (although they could have been controlled by more than one

mutation) and were assigned to one of the eight combinations of basal change (invariable/Orthoplasy/Paraplasy) and plastic change (invariable/Magnification/Mitigation): Paraplasy Magnification, Paraplasy Mitigation, Paraplasy only, Orthoplasy Magnification, Orthoplasy Mitigation, Orthoplasy only, Magnification only, and Migation only (Supplementary Table 7). Genes with cis-acting variants that were shared in both comparisons of *A. halleri* or *A. lyrata* were separated from the lineage-specific derived cis-acting variants, because the evolutionary direction of their effect could not be determined. A chi-squared test with seven degrees of freedom was performed to test whether the proportion of derived cis-acting variants in one of the lineages was different from either the distribution expected based on the other lineage.

**Reporting summary**. Further information on research design is available in the Nature Research Reporting Summary linked to this article.

## Data availability

The RNAseq data generated in this study have been deposited in the Sequence Read Archive (SRA) database under accession code PRJNA640858. The Transcriptome Shotgun Assembly of *Arabidopsis halleri* has been deposited at DDBJ/EMBL/GenBank under the accession GJCZ00000000. The version described in this paper is the first version, GJCZ01000000. The phenotypic data generated in this study are provided in the Supplementary Information.

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

## Acknowledgements

This work was funded by DFG within the CRC680 program, and ERC with Consolidator Grant 648617 "Adaptoscope". We thank Emily Wheeler, Boston for editorial assistance.

## Author contributions

J.D.M., P.K., A.B., F.H. and K.A.S. designed the study. F.H. and M.B. produced the data. F.H., K.A.S., V.K., U.G., M.B. and J.D.M. analyzed the data. K.A.S., F.H. and J.D.M. wrote the manuscript with input from all authors.

## Funding

## Competing interests

The authors declare no competing interests.
