## [Peer Review File · Nature Communications]

REVIEWER COMMENTS

Reviewer #1 (Remarks to the Author):

In this study from He et al., the authors studied how gene expression changes induced by the environment (i.e. expression plasticity) could shape regulatory evolution. To address this fundamental question, they performed carefully controlled RNA-seq assays on two sister species, an outgroup species and their hybrids at six time points before and after induction of dehydration stress. The results are exclusively based on the analysis of RNA-seq data (not validated using other techniques such as RT-qPCR), but the experiment is carefully designed with appropriate controls and replication, resulting in data of high and consistent quality among all samples. The authors reported several interesting trends that help understand how expression plasticity and genetic assimilation can contribute to regulatory evolution in general and to cis-regulatory evolution in particular since allele specific expression (and allele specific plasticity) could be quantified from the hybrids. These findings are original, important and should be of interest to a broad readership since they address long-lasting questions that are not limited to the study system. However, I think in its current state the manuscript is hardly accessible to non-specialists and difficult to evaluate due to the lack of clarity on key methodological points (I really do not understand the y-axis of Fig. 1F-G). I tried to give more details and suggestions below on these important problems that hopefully should not take much work to fix.

Main comments:

I have a couple suggestions for improving the accessibility of the introduction to a broader readership. First, I was confused with the use of "plastic state" in the first paragraph (a term that is not used anywhere else in the manuscript) and I would rather recommend using something like "phenotypic state reached in the novel environment" or "the new phenotypic state". This would emphasize the fact that the context of the study is the adaptation to a novel environment and not the adaptation to dynamic environmental conditions (in which case it is not possible to define a "basal" environment or a "basal" state, but plasticity could very well be adaptive). I think this is important to clear up any potential confusion between these two scenarios at the beginning of the introduction. My second suggestion would be to clearly define "cis-acting modifications" since this is a key concept in the study. This could be done for instance in the second paragraph by explaining that the hypothesis of a single master regulator (mentioned line 54) can be rejected by the identification of cis-acting changes.

Another suggestion to improve the accessibility would be to include in a main figure some of the results currently shown in supplementary figures. In particular, I think it would help the readers to see the experimental design and an overview of basic patterns observed in the RNA-seq dataset before diving into the more complicated and original analyses that are currently shown in Figure 1. For instance, the authors could make a new Figure 1 that would include what is currently shown in Figure S1A and S4A-E. Indeed, there are currently no main figure cited in the first paragraph of the results, which I felt was missing.

Another quick advice to improve clarity would be to add in the result section a sentence similar to lines 102-106 of the Methods. I found that this sentence clearly explained how fundamental properties (genetic variation, plasticity and genetic variation in plasticity) could be analyzed and approached from the data collected by the authors.

Figures are not shown in the order they are cited in the main text, which can be confusing for readers who look at figures after reading a paragraph. I understand that the authors grouped together figures that looked similar, but it is not really helpful or necessary since there is no direct comparison between panels that are side by side (panels 1E-H are shown next to 1A-D, but they are mentioned after 2A in the main text and are not directly compared to 1E-H). Similarly, on figure 4 the panels are not displayed in the order they are cited in the text. I suggest either re-organizing the figures or re-organizing the text.

My main criticism (or mis-understanding) is that the definition of the two classes of genes with mitigated versus magnified response is not clearly explained, making the critical interpretation of

key figures difficult. First, the authors show four clear examples on Fig 1E to illustrate the mitigation or magnification of expression plasticity. However, there are other possible cases that are not covered by these four examples. It is not clear which of these other cases were considered as "Mitigation" or "Magnification" or excluded by the authors. For instance, the expression plasticity may be in opposite directions in the two species or the basal expression may be different between the two species. That would be helpful to show on Fig. 1E how all these cases were classified based on the measures used in the two axes of Fig. 1F-G. Indeed, it is not obvious how the y-axis of Figure 1F-G relates to the four classes shown in sketches of Figure 1E. From what I understand, the measure shown on the y-axis of Figure 1F quantifies whether the expression ratio between the tested species and *A. thaliana* was greater after than before stress. These ratios could have been shown on Figure 1E to make the relation between 1E and 1F more explicit. Also, the abbreviations used in the formulas should be explained in the legend and should avoid mathematical signs such as "-". In addition, there seem to be a mistake in the formula shown on the y-axis of Figure 1F. It reads as $\log_2(x/\log_2(y))$, which means that if the y ratio is below 1, then the result is undefined (\log_2 of a negative value). Adding more to the confusion, the authors mentioned in the Methods (lines 183-186) different ratios for Fig. 1E-G than the ratios actually shown on the y-axis of these figures. For these reasons, I was unable to properly assess whether the two classes of genes shown on Fig 1E-G and used in the rest of the manuscript were meaningful.

The main originality of this work is to link cis-regulatory evolution with expression plasticity evolution. Previous studies already compared allele specific expression in different species and among several environments, although for slightly different purposes. I think this body of work should be cited somewhere in the main text. Suggestions of citations would be: Tirosh et al. Science 2009 (doi:10.1126/science.1169766), Cubillos et al. Plant Cell 2014 (doi:10.1105/tpc.114.130310), Moyerbrailean et al. Genome Res 2016 (doi:10.1101/gr.209759.116), Haas et al. J Exp Bot 2020 (doi:10.1093/jxb/eraa036). Another relevant paper to cite would be Tirosh et al. PNAS 2011 (doi:10.1073/pnas.1113718108).

The sentence at lines 109-110 of the Methods seems to address an important point, but this sentence is not clear enough. How were gene expression levels standardized exactly? More details are needed for the readers to understand how variation in expression plasticity was quantified.

The authors referred to a previous publication in the Methods for the description of allele specific expression quantifications (line 120). Given the importance of these quantifications, they could give a quick description of the approach in addition to the reference, at least to explain the statistical approach used to identify cis-acting expression changes.

Minor points:

Figures 1D and 1H only include a subset of genes. The figure legend should indicate which genes are represented.

For figure A, the main text mentions 9 classes of genes, but the figure only shows 8 classes (no expression changes is missing). Conversely, the legend of Figure S14 mentions 8 classes but the Step 5 on the figure shows 9 classes. Those numbers need to be consistent and if one class is ignored, then it should be mentioned in the text.

The text is too small on Figure 3A.

If the length of branches represents genetic distances on Figure 3B, then the authors should add a scale on the figure.

It is not clear what the stars are showing on Figure 3C.

Line 179, replace "altered significantly" with "were significantly altered".

Line 158, "different" would make more sense than "steeper".

Reviewer #2 (Remarks to the Author):

This manuscript aims to investigate the role of ancestral plasticity in adaptive differentiation among three closely related species of *Arabidopsis* examining expression (including allele specific expression to tease apart trans- v.s. cis- regulation) in response to drought treatments. This is a fascinating question (and one that is difficult to address) that was well motivated in the introduction. Indeed the manuscript was generally very well written and easy to follow. That said, I found that some information relevant to evaluating the approaches could have been presented more clearly and that some necessary caveats were missing from the interpretation of the results. I outline my main questions/concerns below.

My biggest concern is how confidently can we infer ancestral plasticity given that the divergence time between these species is thought to be several millions of years? Not only have patterns of temperature and precipitation changed dramatically over this time period, but the number of mutations since the most recent common ancestor must be several hundred thousand. Additionally, these species have very different life histories (two perennials and one annual) and may thus have evolved responses to drought in very different ways (tolerance versus avoidance/escape) depending on the timing of drought in nature. Yet another consideration is how intraspecific variation in expression plasticity (which I think should be considerable for *A. thaliana* under drought and cold) could affect the interpretation of both ancestral plasticity, differences in plasticity between species. It is my feeling that the assignment of ancestral plasticity in this system is less clear cut than in some of the other cited examples, where there is much more recent divergence and in which they explicitly considered a novel environment in nature. I would suggest incorporating some more discussion of these potential issues into the manuscript proper, and to temper the discussion of support for genetic assimilation (lines 220-224) with these caveats.

It seems to me that timing of the drought treatment (both in developmental stage, and in the onset of the drought treatment) are likely to be important for adaptive differentiation, so I think the essential details of the drought treatment should be described and justified in the manuscript (not just the supplementary methods).

Consider adding some discussion of predictability of the cue that induces plasticity and well as some discussion of the potential costs of plasticity. These seem to bear on whether or not we would expect the Baldwin effect (e.g. lines 39-43), and on the interpretation of paraplasy vs. orthoplasy (e.g. Figure 1a). Granted the costs of plasticity have been notoriously difficult to demonstrate, but there is some evidence that induced responses can be costly. In the case of costly expression and a reliable cue, I think we would not expect orthoplasy (or paraplasy).

I was also somewhat confused by the idea of mitigation (e.g., figure 1e). I can certainly see how homeostasis for fitness would be favorable, but how can we expect an evolved reduction in a (presumably) adaptively plastic response in expression? The stress is still there and needs to be dealt with in some way, correct? Please consider clarifying.

Minor suggestions:

Lines 33-35 – Consider revising, I would say that if plasticity promotes increased survival then it is adaptive rather than the other way around.

Figures S5 – The magnification and mitigation results seems particularly sensitive (lines cross sometimes multiple times). Any suggestions for why that might be, or for what it means for interpreting the results from a given time point?

Supplemental methods - Presumably there is not a space constraint on these methods so I would suggest describing them in full (or at least summarizing them) rather than just a reference to a previous paper.

REVIEWER COMMENTS

Reviewer #1 (Remarks to the Author):

In this study from He et al., the authors studied how gene expression changes induced by the environment (i.e. expression plasticity) could shape regulatory evolution. To address this fundamental question, they performed carefully controlled RNA-seq assays on two sister species, an outgroup species and their hybrids at six time points before and after induction of dehydration stress. The results are exclusively based on the analysis of RNA-seq data (not validated using other techniques such as RT-qPCR), but the experiment is carefully designed with appropriate controls and replication, resulting in data of high and consistent quality among all samples. The authors reported several interesting trends that help understand how expression plasticity and genetic assimilation can contribute to regulatory evolution in general and to cis-regulatory evolution in particular since allele specific expression (and allele specific plasticity) could be quantified from the hybrids. These findings

are original, important and should be of interest to a broad readership since they address long-lasting questions that are not limited to the study system.

We thank the reviewer for this accurate summary of the study.

However, I think in its current state the manuscript is hardly accessible to non-specialists and difficult to evaluate due to the lack of clarity on key methodological points (I really do not understand the y-axis of Fig. 1F-G).

We have noticed a typo on the annotation of the y axis in this figure, which lead to confusion. In addition, the comments of the reviewer helped us understand where confusion arose. In particular, we have deeply reorganized the figures. We hope the present version will have significantly gained in clarity.

I tried to give more details and suggestions below on these important problems that hopefully should not take much work to fix.

We thank the reviewers for these comments, which crucially helped improve the clarity of our manuscript.

Main comments:

I have a couple suggestions for improving the accessibility of the introduction to a broader readership. First, I was confused with the use of “plastic state” in the first paragraph (a term that is not used anywhere else in the manuscript) and I would rather recommend using something like “phenotypic state reached in the novel environment” or “the new phenotypic state”. This would emphasize the fact that the context of the study is the adaptation to a novel environment and not the adaptation to dynamic environmental conditions (in which case it is not possible to define a “basal” environment or a “basal” state, but plasticity could very well be adaptive). I think this is important to clear up any potential confusion between these two scenarios at the beginning of the introduction.

We thank the reviewer for this comment. It is true that the term “plastic state” is confusing and not used any further. We now write (line 33):

“If the phenotypic state reached after activation of a plastic reaction promotes survival in a new environment, then plasticity is adaptive under these conditions. “

My second suggestion would be to clearly define “cis-acting modifications” since this is a key concept in the study. This could be done for instance in the second paragraph by explaining that the hypothesis of a single master regulator (mentioned line 54) can be rejected by the identification of cis-acting changes.

Line 57: we now write “*The study of allele-specific expression in F1 hybrids, instead, offers a powerful avenue to identify myriads of cis-regulatory variants that evolved independently at each transcribed locus. With such a population of variants, it becomes possible to compare rates of evolution among lineages and distinguish the action of natural selection from the random accumulation of new regulatory mutations. The study of cis-regulatory divergence can therefore help explore the general rules governing the evolution of transcriptional plasticity*”.

Another suggestion to improve the accessibility would be to include in a main figure some of the results currently shown in supplementary figures. In particular, I think it would help the readers to see the experimental design and an overview of basic patterns observed in the RNA-seq dataset before diving into the more complicated and original analyses that are currently shown in Figure 1. For instance, the authors could make a new Figure 1 that would include what is currently shown in Figure S1A and S4A-E. Indeed, there are currently no main figure cited in the first paragraph of the results, which I felt was missing.

We appreciate this suggestion of the reviewer. We were happy to follow this advice and reintegrated Figure S1 and S4 into the main manuscript (now Fig. 1). We have reworked the legend and also tried to improve further the clarity of graph annotations.

Another quick advice to improve clarity would be to add in the result section a sentence similar to lines 102-106 of the Methods. I found that this sentence clearly explained how fundamental properties (genetic variation, plasticity and genetic variation in plasticity) could be analyzed and approached from the data collected by the authors.

In the previous version of the manuscript, we privileged brevity. This comment made us realize that in doing so, the reader was missing a summary of the experimental protocol. We have added a short paragraph (line 83-92).

Figures are not shown in the order they are cited in the main text, which can be confusing for readers who look at figures after reading a paragraph. I understand that the authors grouped together figures that looked similar, but it is not really helpful or necessary since there is no direct comparison between panels that are side by side (panels 1E-H are shown next to 1A-D, but they are mentioned after 2A in the main text and are not directly compared to 1E-H). Similarly, on figure 4 the panels are not displayed in the order they are cited in the text. I suggest either re-organizing the figures or re-organizing the text.

We have now replaced what was formally fig 1 and fig2 in two figures: one for basal expression changes and one for plasticity changes. We believe this will address the concern of the reviewer.

My main criticism (or mis-understanding) is that the definition of the two classes of genes with mitigated versus magnified response is not clearly explained, making the critical interpretation of key figures difficult. First, the authors show four clear examples on Fig 1E to illustrate the mitigation or magnification of expression plasticity. However, there are other possible cases that are not covered by these four examples. It is not clear which of these other cases were considered as “Mitigation” or “Magnification” or excluded by the authors. For instance, the expression plasticity may be in opposite directions in the two species or the basal expression may be different between the two species.

That would be helpful to show on Fig. 1E how all these cases were classified based on the measures used in the two axes of Fig. 1F-G. Indeed, it is not obvious how the y-axis of Figure 1F-G relates to the four classes shown in sketches of Figure 1E.

This remark of the review showed us how we had to rewrite the text so that it is clear that all possible cases were examined in our comparison of plasticity reactions among species. We have modified the text in the methods (Line 83-97 and in the methods section line 478-492) to clarify how changes in plasticity were quantified. We also clearly explain now that the “mitigation” category includes genes whose response is smaller in magnitude as well as gene whose direction of expression change is reverse. We note that these genes do not form the bulk of the signal we report, so we preferred not to modify the figure, to avoid complicating it (we still have to comply with the word limit).

From what I understand, the measure shown on the y-axis of Figure 1F quantifies whether the expression ratio between the tested species and *A. thaliana* was greater after than before stress.

This is correct. In other words, the y-axis shows the log of the ratio of the slope in the tested species over the slope of the outgroup (*A. thaliana*). A positive value is obtained when the ratio is greater than 1, that is the tested species responds more than the outgroup. A negative value is obtained when the tested species responds less, or responds in a direction that is opposite to the direction observed in the outgroup (now mentioned on line 180-190 and explained on line 486-490 in methods section).

These ratios could have been shown on Figure 1E to make the relation between 1E and 1F more explicit. Also, the abbreviations used in the formulas should be explained in the legend and should avoid mathematical signs such as “-“. In addition, there seem to be a mistake in the formula shown on the y-axis of Figure 1F. It reads as $\log_2(x/\log_2(y))$, which means that if the y ratio is below 1, then the result is undefined (\log_2 of a negative value). Adding more to the confusion, the authors mentioned in the Methods (lines 183-186) different ratios for Fig. 1E-G than the ratios actually shown on the y-axis of these figures. For these reasons, I was unable to properly assess whether the two classes of genes shown on Fig 1E-G and used in the rest of the manuscript were meaningful.

We apologize for this typo, which is now fixed.

The main originality of this work is to link cis-regulatory evolution with expression plasticity evolution. Previous studies already compared allele specific expression in different species and among several environments, although for slightly different purposes. I think this body of work should be cited somewhere in the main text.

Suggestions of citations would be: Tirosh et al. Science 2009 (doi:10.1126/science.1169766), Cubillos et al. Plant Cell 2014 (doi:10.1105/tpc.114.130310), Moyerbrailean et al. Genome Res 2016 (doi:10.1101/gr.209759.116), Haas et al. J Exp Bot 2020 (doi:10.1093/jxb/eraa036). Another relevant paper to cite would be Tirosh et al. PNAS 2011 (doi:10.1073/pnas.1113718108).

We thank the reviewer for prompting us to re-examine the relevance of these previous findings for the present study. Since we now explain in greater detail the advantage of studying cis-regulatory variation, citing these papers seems a logical addition to the manuscript. We note that given the constraints on manuscript length, we have to be parsimonious on the number of citations, so we focused on the ones reporting patterns of genome-wide cis-acting diversity.

The sentence at lines 109-110 of the Methods seems to address an important point, but this sentence is not clear enough. How were gene expression levels standardized exactly? More details are needed for the readers to understand how variation in expression plasticity was quantified. The authors referred to a previous publication in the Methods for the description of allele specific expression quantifications (line 120). Given the importance of these quantifications, they could give a quick description of the approach in addition to the reference, at least to explain the statistical approach used to identify cis-acting expression changes.

This paragraph in the supplementary has been expanded and now explains the stringent approach we used here to call significant ASE, i.e. by comparing allelic ratios in the transcriptome to a null expectation derived from the measurement of ASE in DNA sequence reads of the hybrid (line 130 in the supplementary methods).

Minor points:

Figures 1D and 1H only include a subset of genes. The figure legend should indicate which genes are represented.

As mentioned above, this is not the case. We hope the reviewer finds it clearer now.

For figure A, the main text mentions 9 classes of genes, but the figure only shows 8 classes (no expression changes is missing). Conversely, the legend of Figure S14 mentions 8 classes but the Step 5 on the figure shows 9 classes. Those numbers need to be consistent and if one class is ignored, then it should be mentioned in the text.

The reviewer is right. It is clearer not to mention the 9th class, which is only needed to compute expected distributions and assign pvalues. We now only mention 8 modes of

plasticity evolution.

The text is too small on Figure 3A.

We increased the size of the text, as well as the size of the legend describing the color code. If the length of branches represents genetic distances on Figure 3B, then the authors should add a scale on the figure.

We agree, but here the genetic distance is not particularly important. The length of the branches reflect the fact that *lyrata* and *halleri* are sister species.

It is not clear what the stars are showing on Figure 3C.

Now given in the legend.

Line 179, replace “altered significantly” with “were significantly altered”.

Changed accordingly.

Line 158, “different” would make more sense than “steeper”.

Changed accordingly.

Reviewer #2 (Remarks to the Author):

This manuscript aims to investigate the role of ancestral plasticity in adaptive differentiation among three closely related species of *Arabidopsis* examining expression (including allele specific expression to tease apart trans- v.s. cis- regulation) in response to drought treatments. This is a fascinating question (and one that is difficult to address) that was well motivated in the introduction. Indeed the manuscript was generally very well written and easy to follow. That said, I found that some information relevant to evaluating the approaches could have been presented more clearly and that some necessary caveats were missing from the interpretation of the results. I outline my main questions/concerns below.

My biggest concern is how confidently can we infer ancestral plasticity given that the divergence time between these species is thought to be several millions of years? Not only have patterns of temperature and precipitation changed dramatically over this time period, but the number of mutations since the most recent common ancestor must be several hundred thousand.

We agree with the reviewers that the species have separated a while ago, and indeed their response to stress differs in many ways. We have also shown that their strategies to respond to drought stress differ – see previous work by Bouzid et al. 2019 in *Annals of Botany*. Yet, the direction of their reaction to stress is very strongly conserved. We highlight this better now in the revised manuscript, because figure 1 shows the correlation of the response of the three species. Most genes that go up, go up in all three species, and most genes that go down, go down in all three species (See Fig 1E, F, G). We can thus confidently assess how genetic change correlates with the direction of the expression response.

Additionally, these species have very different life histories (two perennials and one annual) and may thus have evolved responses to drought in very different ways (tolerance versus avoidance/escape) depending on the timing of drought in nature.

The reviewer is right. Species could differ in their developmental stage and thus express different genes. This lag would cause differences between species in the “trans-“ regulatory environment in which they stand at the time of harvest. It is exactly for this reason that we must include hybrids. The cis-regulatory differences expressed in hybrids confirm that the conclusion we draw from the analysis of the parent is not due to developmental differences. We make this clear now on **line 157**.

Yet another consideration is how intraspecific variation in expression plasticity (which I think should be considerable for *A. thaliana* under drought and cold) could affect the interpretation of both ancestral plasticity, differences in plasticity between species. It is my feeling that the assignment of ancestral plasticity in this system is less clear cut than in some of the other cited examples, where there is much more recent divergence and in which they explicitly considered a novel environment in nature. I would suggest incorporating some more discussion of these potential issues into the manuscript proper, and to temper the discussion of support for genetic assimilation (lines 220-224) with these caveats.

The referee mentions two major difficulties in the study of plasticity evolution between lineages, which we believe address convincingly with our design. Point 1: what was the ancestral plastic state? Point 2: are the genotypes used representative of the species?

Point 1: In experimental evolution, the ancestral population is known. There is no ambiguity on the ancestral plastic state. Such approaches, however, are limited to the study of short evolutionary time scales. Here, we interrogate the basis of plasticity evolution after more than a million year of evolution, ancestral plasticity can therefore only be inferred. In the manuscript, we propose *A. thaliana* as a proxy, because it is an outgroup that is equally distant in evolution from *A. lyrata* and *A. halleri* and because this species is also the one that is used for determining the cis-acting basis of regulatory change. **However, since at least 90% of the genes that respond to stress in *A. thaliana* also respond in the same direction in the two other species, this choice has little impact on our results (see Fig. 1E-G and Fig. S4).**

We now write: line 148 *“To investigate the effect of the proxy on the results, we used the reaction of *A. lyrata* or *A. halleri* as alternative proxy for ancestral plasticity (Fig. S4A-F). The outcome was essentially unchanged, as expected since the direction of the stress response was largely conserved across species.”* And line 195 *“The predominant evolution of a mitigated response to stress in *A. halleri* and the excess of genes showing a magnified response in *A. lyrata* were also observed if we changed the species whose reaction was used as a proxy for the magnitude of ancestral plasticity (Fig. S4E-H).”*

Point 2: The reviewer is right that there is certainly genetic variation within species for the reaction to drought stress. Note that based on previous work, we know that the genotypes we chose are representative of the drought reactions observed in each species (now stated clearly on line 372 in methods section). However, most importantly, our work is based on the analysis of a population of thousands of cis-regulatory modifications, which evolved independently at each locus, and therefore reflect the pattern of cis-regulatory modifications accumulated in each species. Only a subset of these variants will differ between individuals within species.

It seems to me that timing of the drought treatment (both in developmental stage, and in the onset of the drought treatment) are likely to be important for adaptive differentiation, so I think the essential details of the drought treatment should be described and justified in the manuscript (not just the supplementary methods).

We wished we could follow the advice of the reviewer, however, the word limitations forces us to reduce experimental details to the maximum in the main text. However, we would like to point out that what the reviewer describes (impact of developmental stage, perenniality, drought treatment) reflect potential differences in the “trans” regulatory environments. It is

for this reason that including variation in allele-specific expression within hybrid is essential to understand the evolution of the stress response.

We have underscored this aspect in the main text on line 156 (“*This pattern could plausibly result from a single genetic change, if e.g. species were at different developmental stages or if they differed for only one major regulator*”) and line 201 (“*As mentioned earlier, interspecific differences in the slope of the reaction to stress could plausibly result from differences in development or from a handful of potentially random changes in regulators of the stress response.*”)

Consider adding some discussion of predictability of the cue that induces plasticity as well as some discussion of the potential costs of plasticity. These seem to bear on whether or not we would expect the Baldwin effect (e.g. lines 39-43), and on the interpretation of paraplasy vs. orthoplasy (e.g. Figure 1a). Granted the costs of plasticity have been notoriously difficult to demonstrate, but there is some evidence that induced responses can be costly. In the case of costly expression and a reliable cue, I think we would not expect orthoplasy (or paraplasy). Our design includes two controls of variation in cue perception. First, we verified that water loss occurred at the same rate in all three species in our experiment (Fig. S1A). Second, we performed a time serie, which allowed us to spot the time point were the species differed most strongly.

Regarding the cost of plasticity, we did not mention it explicitly in the previous version, but discussed that the pattern seen in *A. halleri* suggests that the necessity for this species to prioritize competitive ability may imply a cost on the stress response (line 69). To address this aspect better, we have now revised the text (line 336), citing the paper by Auld et al. (2010), which is a reference on the topic of plasticity costs:

“Future studies will have to examine whether the excess of cis-acting changes leading to a decrease in plasticity in A. halleri reflects selection to lower the energetic cost associated with plasticity or whether it is a result of relaxed constraints on the regulation of the stress response (Auld et al. 2010)”.

I was also somewhat confused by the idea of mitigation (e.g., figure 1e). I can certainly see how homeostasis for fitness would be favorable, but how can we expect an evolved reduction in a (presumably) adaptively plastic response in expression? The stress is still there and needs to be dealt with in some way, correct? Please consider clarifying.

There are three answers to this question. First, if orthoplastic changes predominate, the slope of the response can be decreased so that the same expression level will be reached (faster) under stress conditions. Second, not all reactions in the stress reactions are instrumental for restoring homeostasis. Some may simply manifest the stress in which the cell is. Adaptation to stress may occur via minimizing such pure stress manifestation. Third, as mentioned above, reduction in the response may be adaptive if responding to stress is costly and growth must be prioritized over survival as might be the case in *A. halleri*. Our manuscript does not allow disentangling between these alternative possibilities, but genetic analyses will hopefully help us clarify this aspect in the future. For the sake of brevity, we keep these questions for a follow-up study that we are conducting in the lab.

Minor suggestions:

Lines 33-35 – Consider revising, I would say that if plasticity promotes increased survival then it is adaptive rather than the other way around.

The reviewer is correct. We changed the text accordingly.

Figures S5 – The magnification and mitigation results seems particularly sensitive (lines cross sometimes multiple times). Any suggestions for why that might be, or for what it means for interpreting the results from a given time point?

We are not sure we understand the question of the reviewer since the panels of the figure describe different aspects of the dynamics of how the two species differ from the outgroup over time. Figure S5-E-G-H show that the excess of genes that evolved a magnified response is specific to *A. lyrata*, particularly pronounced after 6h, but no longer observable after 24h. Fig. S5-H further shows that the proportion of regulatory differences based on cis-modifications vs trans- only changes as the response is deployed.

Supplemental methods - Presumably there is not a space constraint on these methods so I would suggest describing them in full (or at least summarizing them) rather than just a reference to a previous paper.

Modified accordingly (see line **490-507 and 566-575** in suppl. Methods).

REVIEWERS' COMMENTS

Reviewer #1 (Remarks to the Author):

In my opinion, the authors convincingly addressed the main issues raised by the reviewers in the revised version of their manuscript entitled "Cis-regulatory evolution spotlights species differences in the adaptive potential of gene expression plasticity". In particular, they added essential and clearly written explanations on how and why experiments and statistical analyses were performed and they re-organized the figures in a more intuitive order that better follows the text. I think these changes will significantly improve the accessibility of the results and of their importance for non-specialist readers. I do not have major concerns about the revised manuscript, only two minor suggestions:

- Line 225, the authors wrote that a chi-squared test was used to show that genes were not uniformly distributed among 8 different categories. My understanding is that "uniformly distributed" refers to the expected distribution shown on Figure 4A. However, it is not obvious how the expected number of genes was obtained for the 8 categories and why it is called "uniformly distributed". I could not find any explanation in the Methods. Although in general the Methods are very clear, I think the statistical analyses used on Figure 4A still deserve further explanation.

- Line 59, the authors wrote that they identified "cis-regulatory variants that evolved independently at each transcribed locus". In my understanding, the cis-regulatory variants arose independently during evolution, but then they might not have necessarily evolved independently from each other (for instance in case of epistatic effects on fitness or due to demographic effects). I would therefore suggest writing "arose independently during evolution" rather than "evolved independently".